# Transcriptomic and Metabolomic Analyses of the Response of Resistant Peanut Seeds to *Aspergillus flavus* Infection

**DOI:** 10.3390/toxins15070414

**Published:** 2023-06-26

**Authors:** Yun Wang, Dongmei Liu, Haiyan Yin, Hongqi Wang, Cheng Cao, Junyan Wang, Jia Zheng, Jihong Liu

**Affiliations:** Institute of Agricultural Quality Standards and Testing Technology, Henan Academy of Agricultural Sciences, Zhengzhou 450002, China; wangyun1906@163.com (Y.W.); dongmeiliu80@163.com (D.L.); yinhaiyan2005@126.com (H.Y.); huda2000@126.com (H.W.); lllaohucao@163.com (C.C.); wjyan1735@163.com (J.W.); zhengjia8909@163.com (J.Z.)

**Keywords:** peanut, transcriptomic, metabolomic, aflatoxins, *Aspergillus flavus*

## Abstract

Peanut seeds are susceptible to *Aspergillus flavus* infection, which has a severe impact on the peanut industry and human health. However, the molecular mechanism underlying this defense remains poorly understood. The aim of this study was to analyze the changes in differentially expressed genes (DEGs) and differential metabolites during *A. flavus* infection between Zhonghua 6 and Yuanza 9102 by transcriptomic and metabolomic analysis. A total of 5768 DEGs were detected in the transcriptomic study. Further functional analysis showed that some DEGs were significantly enriched in pectinase catabolism, hydrogen peroxide decomposition and cell wall tissues of resistant varieties at the early stage of infection, while these genes were differentially enriched in the middle and late stages of infection in the nonresponsive variety Yuanza 9102. Some DEGs, such as those encoding transcription factors, disease course-related proteins, peroxidase (POD), chitinase and phenylalanine ammonialyase (PAL), were highly expressed in the infection stage. Metabolomic analysis yielded 349 differential metabolites. Resveratrol, cinnamic acid, coumaric acid, ferulic acid in phenylalanine metabolism and 13S-HPODE in the linolenic acid metabolism pathway play major and active roles in peanut resistance to A. flavus. Combined analysis of the differential metabolites and DEGs showed that they were mainly enriched in phenylpropane metabolism and the linolenic acid metabolism pathway. Transcriptomic and metabolomic analyses further confirmed that peanuts infected with *A. flavus* activates various defense mechanisms, and the response to *A. flavus* is more rapid in resistant materials. These results can be used to further elucidate the molecular mechanism of peanut resistance to *A. flavus* infection and provide directions for early detection of infection and for breeding peanut varieties resistant to aflatoxin contamination.

## 1. Introduction

Peanuts, *Arachis hypogaea* L., enriched in oil, protein and other nutrients is an important cash crop and plays a key role in the edible vegetable oil and leisure food industries [1,2]. However, because of its high fat and protein content and nutritional characteristics, peanuts are prone to be infected by *Aspergillus flavus* and mildewed under the combined action of high temperature, high humidity, oxygen, sunlight, mechanical damage and microorganisms, resulting in aflatoxin contamination [3]. Aflatoxin contamination occurs not only in pre-harvest but also in postharvest. Compared with pre-harvest, aflatoxin contamination is particularly severe during storage, especially in areas and years with continuous rain during the peanut harvesting period and peanut aflatoxin levels exceeding the standard are often observed. The contamination subsequently leads to compromise the quality and endanger the safety of peanut consumption as aflatoxins are the most toxic and carcinogenic compounds among the toxins [4,5]. To date, lots of biological strategies to battle contamination have been developed with limited success [6,7,8]. Thus, it is an effective approach to breed the peanut-resistant cultivars for controlling aflatoxin contamination. while the progress is slow, due to lack of understanding the underlying resistance mechanism [9].

Aflatoxin production is related to a variety of factors, such as peanut seed maturity, water activity, storage conditions and resistance of peanut seeds, while temperature and water activity are the most important environmental factors [10]. The resistance of peanut seeds to pathogen is an active defense process [11,12]. During the process of *A. flavus* infection of peanuts, some physiological and biochemical indicators change, such as phenylalanine ammonia lyase (PAL) [13], peroxisome (POD) [14] and lipoxygenase (LOX) [15,16] activities. These defense enzymes, which participate in the synthesis of lignin, phenolic compounds and plant protectants, help peanut plants resist the damage caused by reactive oxygen species and oxygen free radicals to the cell membrane system and enhance the resistance of peanut plants to *A. flavus* infection [17]. Some resistance-related genes and metabolites are not only involved in the process of resistance to *Aspergillus* infection but also are important active substances to help inhibit or reduce *A. flavus* toxicity in peanut plants. For example, LOX regulates fatty acid metabolism and produces a series of products with signaling functions, which can induce the expression of plant resistance-related genes and improve plant resistance to mechanical damage and pathogen infection [18]. WRKY transcription factors, heat shock proteins, TIR-NBS-LRR and other genes are also important components of resistance to *A. flavus* infection [19]. PR genes play an important role in peanut resistance to *A. flavus* infection and toxicity, including the chitinase gene and PR10 gene [20,21]. The expression levels of related differentially expressed genes (DEGs) in the phenylpropanoid metabolic pathway and the astragalus compound metabolic pathway were found to be significantly different among different resistant peanut varieties [22]. Polyphenols and antibacterial proteins can help plants resist the infection and toxicity of *A. flavus* in peanuts. For example, the content and synthesis rate of resveratrol in peanuts are related to resistance to peanut disease [23].

Omics research methods with the characteristics of high throughput are widely used in the study of plant resistance mechanisms, providing comprehensive information and aiding in further exploration of the molecular mechanism of resistance. Metabolomics and transcriptomic association analysis widespread application in studying many plants [24,25] and some transcriptomic and proteomic studies have been conducted on peanut resistance to *A. flavus* infection [26,27,28], but few studies have applied the combination of peanut postharvest transcriptomics and metabolomics to the study of peanut resistance to *A. flavus* infection.

Therefore, the objectives of this work were to investigate the transcriptomic and metabolomic changes in different resistant peanut cultivars to *A. flavus.* Moreover, we uncovered a potential regulatory network between genes and metabolites that involved in defense response. The outcomes of the present study provide a comprehensive framework for better understanding the potential molecular adaptation strategy of the resistant peanut variety in response to *A. flavus* at the transcriptomic and metabolomic levels, providing insights for breeding disease-resistant varieties.

## 2. Results

### 2.1. Aflatoxin, Resveratrol Content and the Changes in Resistance-Related Enzyme Activities after Inoculation

#### 2.1.1. Changes in Aflatoxin Content in Peanut with Different Water Activity after Inoculation

The results showed that peanut water activity (aw) had a great influence on the growth of *A. flavus*. The growth of *A. flavus* became more vigorous with increasing aw when the aw was between 0.85 and 0.99. As shown in Figure 1, the AFB1 content in peanut after *A. flavus* inoculation showed an upwards trend at different water activities. The suitable water activity range for toxic production was 0.92–0.97, and the optimal water activity for toxic production was 0.97. At aw ≤ 0.75 and aw ≥ 0.99, the growth of *A. flavus* was inhibited.

#### 2.1.2. Dynamic Changes in Resveratrol and Resistance-Related Enzyme (PAL, POD, LOX) Activities after Inoculation with *A. flavus* at a Suitable Water Activity Level

The detection and analysis of the resveratrol content in the two peanut varieties before and after inoculation with *A. flavus* showed that the total resveratrol content peaked at 4440.54 µg/kg in the resistant peanut seeds (Zhonghua 6) after inoculation with *A. flavus* at 1 d, then rapidly reduced, and it eventually disappeared at 7 dpi. The change trend of resveratrol content of Yuanza 9102 after inoculation was basically the same as that of Zhonghua 6. However, the synthesis rate of resveratrol in Yuanza 9102 peanut seeds was Significantly lower than that in Zhonghua 6 after inoculation with *A. flavus*, and the rate in the resistant variety peaked at 1 d (Figure 2).

Compared with the control, the PAL and POD activities increased markedly after inoculation in both varieties (Figure 2). The PAL activity of Zhonghua 6 peaked at 1 dpi and increased 67.68%, while the PAL activity of the Yuanza 9102 variety peaked at 4 dpi. The POD activity of Zhonghua 6 peaked at 2 dpi and increased 52.47%, while the POD activity of the Yuanza 9102 variety peaked at 3 dpi. and increased 32.92%.

The LOX activity of Zhonghua 6 was higher than that of Yuanza 9102 in all stages in the control, but the difference was not significant. The LOX activity of Zhonghua 6 and Yuanza 9102 increased significantly at 1 dpi and 3 dpi, but the LOX activity of Zhonghua 6 increased more than Yuanza 9102. The LOX activity of Zhonghua 6 peaked at 3 dpi increased 60.75% compared to the control, while the LOX activity of the Yuanza 9102 variety peaked at 4 dpi of which value was increased 50.21% (Figure 2). In general, the activities of the three enzymes of Zhonghua 6 and Yuanza 9102 all increased significantly after inoculation.

### 2.2. Transcriptomic and Metabolmic Profiles of A. flavus-Infected Peanut Seeds

Samples of six groups invaded by *A. flavus* and six controls with no infection from different resistant peanut varieties at different times were prepared and analyzed. Results showed that 46.78 Mb to 48.50 Mb of raw reads from the 12 libraries were obtained and the clean read ratios of the sequencing samples were all more than 90% (Appendix A). The average Q20 and Q30 of the clean reads of all libraries were >96% and >90%, respectively, which indicated all samples libraries passed quality standards. Based on the sequencing data, the rates of total mapped clean reads of the Y_YT3 sample averaged 67.26% because the sample was severely infected by *A. flavus* at this time.

In order to compare the gene expression differences between different resistant varieties at different times, the samples were set into six groups, including Y-T1 vs. Y-CK1(1 dpi), Y-T2 vs. Y-CK2(3 dpi), Y-T3 vs. Y-CK3(7 dpi), Z-T1 vs. Z-CK1(1 dpi), Z-T2 vs. Z-CK2(1 dpi), and Z-T3 vs. Z-CK3(1 dpi), which Y is Yuanza 9102 peanut seeds with moderately susceptible to *A. flavus* and Z is Zhonghua 6 peanut seeds and resistant. The gene expression levels were calculated and considered differentially expressed depending on the criteria of adjusted q-value ≤ 0.05 and|log2FoldChange| ≥ 2. A total of 5768 DEGs were detected in the six comparison groups.

Compared with the control, 4615 DEGs were identified in Yuanza 9102 and 4009 DEGs in Zhonghua 6. The amount of upregulated DEGs was higher than that of downregulated genes at each stage of infection in both varieties (Figure 3A). Among the DEGs, the most abundant genes were expressed, including genes of pathogenesis-related proteins (PR), chitinase, β-1,3-glucanase, phenylalanine ammonialyase (PAL), chalcone synthase, stilbene synthase (STS), 4-coumarate: CoA ligase (4CL), peroxidase (POD), transcription factors (ERF, WRKY, MYB, NAC, etc.), and lipoxygenases (LOX). (The relative expression levels of all these DEGs at different inoculation stages of Yuanza 9102 and Zhonghua 6 are presented in Appendix A). PR1, PR2, PR5 of PR proteins were found to be highly expressed during *A. flavus* infection at 1 dpi and 3 dpi. Meanwhile, PR-like proteins such as chitinase and β-1,3-glucanase were also found to be accumulated in both infected samples. The expression levels of PAL, 4CL, CHS and STS, key enzymes involved in phenylpropane metabolism in peanuts, were increased in both peanut varieties after *A. flavus* infection (Appendix A). Peroxidase classes (peroxidase 4, 5, 10, 51, A2 etc.), lignin-forming anionic peroxidases and cationic peroxidases were highly expressed at the early stage of infection of Zhonghua 6 (1 dpi). In addition, WRKY, MYB, ERF and NAC transcription factors were mainly upregulated at 1 dpi and 3 dpi. Six differentially expressed LOX genes (2 9S-LOX and 4 13S-LOX genes) were found, and all of them were upregulated to resist *A. flavus* in stages of infection.

To understand the molecular changes that occurred in different peanut seeds to *A. flavus* infection, metabolomic analysis was also performed. Overall, 349 candidates showed differential metabolites in the infected peanut seeds based on the screening criteria of adjusted q-value ≤ 0.05 and fold change Fold-Change ≥ 2 or ≤0.5. 268 and 238 differential metabolites were screened in the Zhonghua 6 (Z-T vs. Z-CK) and Yuanza 9102 (Y-T vs. Y-CK) comparison groups, respectively (Figure 3C). among which 120 differential metabolites (94 upregulated and 26 downregulated) in the Z-T1 vs. Z-CK1 comparison group. 155 differential metabolites in the Z-T2 vs. Z-CK2 comparison group (132 upregulated and 23 downregulated), 77 differential metabolites in the Z-T3 vs. Z-CK3 comparison group (37 upregulated and 40 downregulated), 139 differential metabolites in the Y-T1 vs. Y-CK1 comparison group (54 upregulated, 85 downregulated), 103 differential metabolites (87 upregulated and 16 downregulated) in the Y-T2 vs. Y-CK2 comparison group, and 70 differential metabolites (40 upregulated and 30 downregulated) in the Y-T3 vs. Y-CK3 comparison group were found (Figure 3B). There were 157 different metabolites in both varieties, which indicated the different varieties had a certain consistency in the response to *A. flavus* infection at the metabolic level (Figure 3C). The classification results for differential metabolites showed that (Figure 3D) these differential metabolites mainly included flavonoids, terpenoids, benzene and its derivatives, polyketides, fatty acyl compounds, phenylpropanoids, and alkaloids.

The expression of alpha-linolenic acid was upregulated at 1 dpi and 3 dpi in Zhonghua 6, but it was upregulated at 3 dpi in Yuanza 9102. 13S-HPODE was upregulated at 1 dpi and 3 dpi in both varieties, and the content of 13 S-HPODE in Zhonghua 6 was higher than that in Yuanza 9102. Trans-Cinnamic acid, Ferulic acid, Sinapic acid and Coumaric acid in the phenylpropane metabolic pathways were upregulated after inoculation. Resveratrol and its derivatives picetotaxol, caffeic acid, daidzein and its downstream product formononetin was upregulated in peanut inoculated with *Aspergillus flavus* at 1 d and 3 d (Appendix A).

### 2.3. Functional Analysis of DEGs and DAMs

To further investigate the biological metabolic process of plant response to pathogen infection at different time points after infection, GO significant enrichment analysis was conducted on DEGs of the two peanut varieties treated at different time points and the control group (Appendix A). Although Yuanza 9102 and Zhonghua 6 had common GO enrichment entries, such as oxidoreductase activity, defense reaction and cell wall, there were some differences genes including the pectin catabolic process, hydrogen peroxide catabolic process and cell wall organization were enriched at the early stages of infection (1 dpi) in Zhonghua 6, while they were enriched at 3 dpi and 7 dpi in Yuanza 9102 (Appendix A).

KEGG pathway significant enrichment analysis was conducted on DEGs of the two peanut varieties treated at different time points (Figure 4). Common resistance metabolic pathways containing monoterpenoid biosynthesis, MAPK signaling pathway-plant, phenylpropane biosynthesis, and flavonoid biosynthesis, were found in the two cultivars, but some metabolic pathways involved in the defense response consisting of cutin, suberine and wax biosynthesis, zeatin biosynthesis, tyrosine metabolism, α-linolenic acid metabolism, isoquinoline alkaloid biosynthesis, stilbene, diarylheptane and gingerol biosynthesis, tropane, piperidine and pyridine alkaloid biosynthesis were expressed earlier in Zhonghua 6.

The result from the metabolism data showed the metabolic pathways which the different metabolites were enriched gradually decreased with increasing infection time. Although the main metabolic pathways were basically the same, there were differences in some metabolic pathways (Figure 5). α-linolenic acid metabolism was enriched throughout the infection period of Zhonghua 6 but was significantly enriched at 3 dpi in Yuanza 9102. The biosynthesis of aminoacyl tRNA and 2-oxycarboxylic acid metabolism were significantly enriched at 1 dpi in Yuanza 9102, while these pathways were enriched at 1 dpi and 3 dpi after inoculation in Zhonghua 6. Tyrosine metabolism was enriched at 1 dpi and 3 dpi in Zhonghua 6 but only at 7 dpi in Yuanza 9102.

### 2.4. qRT–PCR Validation of Selected DEGs

To validate the transcriptomic results, 12 DEGs implicated in plant pathogen defense were quantified by qRT–PCR. Of these, chalcone synthase, chalcone-flavonoid ketone isomerase (CHI), stilbene synthase (STS), phenylalanine ammonialyase (PAL), 4-coumarate: CoA ligase (4CL) and peroxidase were involved in the metabolic pathways of flavonoids and phenylpropane. Chitinase, pectin esterase and cellulose synthase were related to defense reaction and cell wall. Others are ethylene-responsive transcription factor, pathogenesis-related protein and 13S-LOX. As shown in Figure 6, qRT–PCR analyses showed basically the same expression trend for each of the analyzed candidates. These results suggest that the transcriptomic data were accurate and could be used for further functional analysis.

### 2.5. Integrative Analysis of the Transcriptome and Metabolome

Correlation analysis of DEGs and differential metabolites was conducted. The results of KEGG pathway enrichment analysis showed that the metabolic pathways of α−linolenic acid and phenylpropane were significantly enriched. Therefore, transcriptome data and metabolite analysis were integrated to analyze the changes in DEGs and differential metabolites involved in these two pathways. The dynamic changes in DEGs and differentially expressed metabolites in the α–linolenic acid metabolic pathway is shown in Figure 7A. Two genes encoding FAD2, one gene encoding FAD3, and the LOX (lipoxygenase) gene were upregulated in Zhonghua 6 after infection, the differential metabolites such as linoleic acid, alpha-linolenic acid, 13S–HPODE and 9S–HPODE were enriched in the related metabolic pathway. The dynamic change process of DEGs and differentially expressed metabolites in the phenylpropane metabolic pathway were shown in Figure 7B. Three PAL genes, five genes encoding 4CL, four genes encoding CHS and four CHI genes which were detected 1 dpi and 3 dpi were upregulated and in the related metabolic pathway, the differential metabolites 4–hydroxycinnamic acid, coumaric acid, caffeic acid, ferulic acid and erucinic acid were enriched.

## 3. Discussion

The aim of this study was to provide a better understanding of the molecular defense mechanism in peanuts after infected by *A. flavus*. We compared the transcriptomic profiles between infected and uninfected peanut seeds. Meanwhile, we also analyzed the metabolome differences between the two varieties in an attempt to provide corroborating or at least supplementary results to the transcriptomic analysis. GO analysis showed that biological processes as pectin catabolic process, hydrogen peroxide catabolic process and cell wall organization existed significant differences in the two varieties, especially at 1 dpi. KEGG analyses at both the transcriptomic and metabolic levels showed enrichment of the phenylpropane and α-linolenic acid metabolic pathways.

The aflatoxin content of peanuts after harvest was significantly affected by peanut water activity. In this paper, it was found that the suitable water activity of peanut for toxin production was 0.92–0.97, and the A. flavus growth was inhibited when water activity of peanut was under 0.75, which was consistent with Abdel-Hadi research [29]. Thus, peanut plant should be dried in time after harvest to reduce their water activity to under 0.75 in order to inhibit to the growth of A. flavus and the production of aflatoxins.

PR proteins are an indispensable component of innate immune responses in plants under biotic or abiotic stress conditions and have been induced and accumulated in plants in response to these adverse conditions [30,31,32]. The expressions of PR families such as PR1, PR2, PR5 and PR10 were induced to a higher level to trigger the rapid activation of resistance mechanisms after infection in many plants [33,34]. DEGs of PR1, PR2, and PR5 were highly expressed in the initial period of infection(1 dpi) in Zhonghua 6 and Yuanza 9102 (Appendix A). Chitinases and β-1,3-glucanases which were characterized as PRs, can decompose fungal cell walls and are important components of the plant defense response [20,35]. β-1,3-Glucanase usually acts synergistically with chitinase in plant response to pathogen infection, and both enzymes can improve peanut resistance to *A. flavus* infection [36]. The expression levels of chitinase and β-1,3-glucanase genes of Zhonghua 6 and Yuanza 9102 peanuts were also increased to enhance the ability of peanut to resist *A. flavus* (Appendix A). Upregulation of PR proteins is positively regulated by ethylene signalling in the plant response to biological stress [37]. Twenty-two ethylene-responsive transcription factors of Zhonghua 6 and Yuanza 9102 peanuts were found (Appendix A); thus, we predict that they may participate in regulating PR genes. In general, these genes participated in defense response but were not the main reason for resistant differences in Zhonghua 6 and Yuanza 9102.

GO enrichment analysis and KEGG pathway enrichment analysis of specific DEGs in the two peanut varieties showed that plant physical defense pathways (e.g., pectin catabolic process, cutin, suberine and wax biosynthesis) were enriched earlier in resistant varieties. This was consistent with the transcriptome analysis results in peanut tissues during the response to white silk disease [38]. Lipid transfer protein (LTP) and GPI-anchored lipid transfer protein 1 (LTPG/LTPG1) were found to be involved in the accumulation of epidermal wax [39]. The expression levels of the wax, keratin and suberine biosynthesis genes Eceriferum, Hothead gene, and fatty acyl-CoA reductase were significantly higher in Zhonghua 6 than in Yuanza 9102 at the early stage of infection (1 dpi) (Appendix A). The expression of cell wall reconstruction-related genes such as pectinesterase, xyloglucan endotransglucosylase/hydrolase and cellulose synthases was greatly upregulated in the early stage of infection in Yuanza 9102 and Zhonghua 6 (Appendix A). These results indicated that different resistance between Zhonghua 6 and Yuanza 9102 to *A. flavu* may be related to the expression of the structural factors that formed the physical barrier.

PAL is the rate-limiting enzyme of the phenylpropane metabolic pathway in plants. Nandini et al. found that the PAL activity in the phenylpropane metabolic pathway showed an increasing trend after peanut plants were infected with *A. flavus* [40]. Nayak et al. also found that PAL gene expression in resistant varieties occurred earlier and was higher than that in susceptible varieties in a transcriptomic study of the resistance to *A. flavus* infection in peanut [13]. The PAL activity of the resistant materials (Zhonghua 6) increased rapidly and peaked on day 1 dpi, while moderately susceptible materials (Yuanza 9102) peaked at 4 dpi. What is more, the genes expression levels of PAL, 4CL, CHS and STS, key enzymes involved in phenylpropane metabolism in Zhonghua 6, were higher than those in Yuanza 9102 peanut (Appendix A). The key difference between resistant and susceptible plants is the timely recognition of the invading pathogen or stress and the rapid and effective activation of host defense mechanisms [41]. It can be inferred that the PAL activity of Zhonghua 6 to *A. flavus* is activated rapidly. Therefore, PAL activity on 1 dpi is one of the main factors causing the resistance difference between the two varieties.

The phenylpropanoid pathway can produce antibacterial metabolites such as rutin, quercetin, anthocyanin and ferulic acid [42,43]. Cinnamic acid and coumaric acid in this pathway can inhibit P450 monooxygenase activity, which can be used by *A. flavus* to produce aflatoxin [44]. Ferulic acid, coumaric acid and erucic acid played an important role in plant defense responses [23]. In this study, the expression levels of coumaric acid, erucic acid, caffeic acid, and ferulic acid were significantly upregulated after inoculation with *A. flavus* (Appendix A). The resveratrol content measured in Zhonghua 6 to *A. flavus* was significantly higher than that in Yuanza 9102. This is related to Wang’s research on the relationship between resveratrol content and resistance to aflatoxin accumulation caused by *A. flavus* [45].

POD is an important defense enzyme in plants. Changes in POD activity play an important role in the physiological defense of plants to pathogenic bacteria at early invasion stage [46]. When apples [47], grapes [48] and strawberries [49] are infected by pathogenic bacteria, POD enzyme activity enhances fruit resistance. POD activity increased significantly in 1 dpi and 3 dpi. Forty-seven differentially expressed POD genes were identified by transcriptome analysis and one POD gene encoded lignin-forming anionic peroxidase (Appendix A) which can catalyze the oxidation of phenolic substances, generate lignin to form defensive barriers and enhance cell structure [50,51,52].

In addition to genes directly involved in disease resistance, transcription factors, as essential regulatory proteins, are involved in the regulation of gene expression to a large extent. ERF [53], WRKY [54], MYB [55], NAC [56] and other transcription factors can improve the resistance of plants to various biological stresses. The most differentially expressed WRKYs, MYBs and ERFs were mainly upregulated in the process of peanut resistance to *A. flavus*. In addition to directly participating in defense responses, transcription factors such as NAC and MYB can also act as regulatory factors for lignin biosynthesis in plants [56]. The upregulation of NAC and MYB in the early stage of infection may play an important role in Zhonghua 6 to *A. flavus* infection (Appendix A).

LOX and its metabolic derivatives catalyze fatty acid production and play an important role in plant defense [57,58], and the improvement of LOX activity can enhance plant response [18]. When crops are affected by fungal diseases, LOX can be activated to catalyze unsaturated fatty acids to produce the aflatoxin-inhibiting metabolite 13S-HPODE [59]. In this study, the differential metabolites of peanuts were significantly enriched in the biosynthetic pathway of unsaturated fatty acids mainly in α-linolenic acid metabolism. The accumulation of 13S-HPODE was significant at 1 dpi and 3 dpi, and the 13S-HPODE content in Zhonghua 6 was higher than that in Yuanza 9102 (Appendix A). Jasmonic acid (JA) can improve the ability of plants to resist pathogen infection [60]. JA in Zhonghua 6 significantly accumulated on the day of *A. flavus* infection and participated in peanut resistance to *A. flavus* infection.

## 4. Conclusions

Transcriptomics and metabolomics were used to analyze the mechanism underlying the resistance of peanut to *A. flavus* infection. In this study, 5768 DEGs and 349 differentially abundant metabolites were identified. Combined analysis of differential metabolites and DEGs found that they were mainly concentrated in phenylpropane metabolism and the linolenic acid metabolism pathway. Some DEGs, such as those encoding transcription factors, disease course-related proteins, peroxidase (POD), chitinase and phenylalanine ammonialyase (PAL), were highly expressed in the infection stage. Metabolomic analysis yielded 349 differential metabolites. Resveratrol, cinnamic acid, coumaric acid, ferulic acid in phenylalanine metabolism and 13S-HPODE in the linolenic acid metabolism pathway play major and active roles in peanut resistance to *A. flavus*. All these results further confirmed that peanuts infected by *A. flavus* can activate a variety of defense mechanisms, and the response to *A. flavus* is more rapid in Zhonghua 6. These results can be used to further elucidate the molecular mechanism of peanut resistance to *A. flavus* infection and provide directions for early detection of infections and for breeding peanut varieties resistant to aflatoxin contamination.

## 5. Materials and Methods

### 5.1. Peanut Material and Pathogen Inoculation

Common (Yuanza 9102) genotypes which are the moderately susceptible variety and resistant seeds (Zhonghua 6) were provided by Oil Crops Research Institute, Chinese Academy of Agricultural Sciences. Highly toxigenic strain *A. favus* 3.2890 from our lab was cultured on Potato Dextrose Agar for 7 days at 30 °C. Conidia were then collected and suspended in sterile water containing 0.05% Tween-80 with a concentration of 1 × 10^4^ spores/mL and were stored at −70 °C.

Samples were taken continuously for 7 days while the fresh and healthy postharvested peanuts were spread out to dry under natural weather, then shelled peanuts and measured the water activity of samples. At the same time, 20.0 g of the shelled peanuts that were surface sterilized for 45 s by 75% ethanol and rinsed thrice with sterile distilled water were inoculated with 1 mL spore suspension (1 × 10^4^ spores/mL) in a plantlet bottle. In the control, 1 mL 0.05% Tween-80 solution was placed on the peanut seeds. The inoculated samples and the control were placed in a constant-temperature incubator with 100% relative humidity at 30 °C in the dark. Samples were taken continuously for 7 days after inoculation and immediately frozen in liquid nitrogen and stored at −80 °C. The experiment for each group was repeated three times.

### 5.2. Aflatoxin, Enzyme Activity and Resveratrol Content Assay

The content of aflatoxin was detected according to the description by wang et al. [23]. Enzyme activity and resveratrol were measured at the optimal water activity of 0.97. PAL and POD were extracted and detected according to the method from Wang et al. [61]. Resveratrol content was calculated followed by Limmongkon et al. [62].

Lipoxygenase (LOX) was measured by plant lipoxygenase (LOX) Activity Assay Kit (Sangon Biotech, Shanghai, China) and the introductions are as follows: 0.1 g Fresh sample was fully grinded on ice and then centrifuged at 4 °C for 20 min at 16,000 rpm, after which the clear supernatant was extracted and the enzyme activity was determined on ice. Before this experiment, the spectrophotometer was preheated for more than 30 min and adjusted the wavelength to 234 nm by distilled water to zero. As blank tube: 100 μL distilled water, 800 μL reagent I and 100 μL reagent II were quickly added into 1mL quartz colorimetric dish and determined at 234 nm at 15 s and 75 s recording as A1 and A2, respectively. The blank tubes only need to be done 1–2 times. Determination tube: 100 μL supernatant, 800 μL reagent I and 100 μL reagent II were quickly added into 1mL quartz colorimetric dish and determined at 234 nm at 15 s and 75 s recording as A3 and A4, respectively. The activities were determined by equation LOX(U/g) = 1000 × [(A4 − A3) − (A2 − A1)]/W, where W is the fresh weight used.

### 5.3. RNA Extraction and Transcriptome Sequencing

The peanut seeds of Zhonghua 6 and Yuanza 9102 (inoculated with *A. flavus* and the control group) were collected at 1 dpi, 3 dpi and 7 dpi under the optimum water activity for RNA isolation. The total RNA of peanut seeds was extracted using RNeasy Plant Mini Kit (Qiagen, Hilden, Germany). Three replicate samples were extracted from each group. Agarose gel electrophoresis was used to identify the purity of the extracted RNA. A Nanodrop 2000 was used for RNA concentration. The cDNA library was constructed by BGI Genomics. High-throughput transcriptome sequencing was performed with a MGISEQ-2000 sequencer.

### 5.4. Alignment to the Reference Genome and Functional Annotation of DEGs

The raw reads were first filtered to remove low-quality reads, adaptor reads and reads with ambiguous ‘N’ nucleotides (with an ‘N’ ratio over 10%) to obtain high-quality clean reads. The clean reads (Q20 > 96% and Q30 > 90%) were used for further alignment and assembly. The clean data were mapped to the peanut reference genome (NCBI GCF_000817695.2_Aradu1.1) using HISAT (version 2.0.5).

The expression level of a gene was normalized to the fragments per kilobase of transcript per million fragments mapped (FPKM) value. Differential gene expression analysis was performed using the DEGseq method of the R statistical package. We used q-value < 0.05 and log2|fold change| ≥ 2 as thresholds for significantly different expression levels. GO functional enrichment analyses were performed to identify the biological, cellular and molecular functions of DEGs. Similarly, all DEGs were mapped to terms in the KEGG database and GO terms and KEGG pathways with false discovery rate (FDR)-corrected q-value < 0.05 were considered statistically significant.

### 5.5. Validation of Transcriptome Data by RT–qPCR Assays

To validate the expression profiles obtained by RNA-seq, 12 differentially expressed resistance genes were selected to validate their relative expression by RT–qPCR. Specific primers of key genes were designed using Primer Premier software (5.0) (Appendix A). Total RNA was reverse-transcribed into single-stranded cDNA using the Prime Script RT Reagent Kit with gDNA Eraser (Takara, Kusatsu, Japan). The ACTIN gene served as an internal control for normalizing the transcript levels of all analyzed target genes [63]. Reactions were carried out on CFX96 (Bio-Rad, Hercules, CA, USA) using SYBR Premix ExTaq reagents (Takara, Japan) following the manufacturer’s recommended protocol. Three technical replicates were included for each biological replicate, and the relative expression levels of the selected genes were calculated using the 2^−ΔΔCT^ method.

### 5.6. Metabolomic Sequencing Analysis

To study the metabolite variations among different growth periods at 1 dpi, 3 dpi and 7 dpi under the optimum water activity, nontarget metabolic analyses were performed on the samples with three biological duplicates by BGI Genomics. Metabolite extraction: Approximately 50 mg crushed sample was added to a 1.5 mL Eppendorf (EP) tube, then 0.8 mL of extraction liquid (Vmethanol: Vwater = 7:3, −20 °C for pre-cooling) and 20 μL internal standard were added. Afterward, the sample was treated via ultrasound for 30 min at 4 °C; each tube was then centrifuged at 14,000 rpm for 15 min at 4 °C. The supernatant (0.6 mL) was filtered through 0.22 μm filter membrane and subsequently transferred to a clean GC–MS glass vial, of which 20 μL of each sample was removed and pooled as a quality control (QC) sample.

LC–MS/MS analysis: The instrument platform for LC–MS analysis was the Q-Exactive Orbitrap mass analyzer of Termo Fisher Scientific. Chromatographic conditions: 2 μL of the sample was separated by a Hypersil GOLD Q column (100 mm × 2.1 mm, 1.9 μm). The mobile phases consisted of 0.1% formic acid in water (solvent A) and 0.1% formic acid in acetonitrile (solvent B). The chromatographic gradient elution procedure is 0~2 min, 5% solvent B; 2~22 min, 5%~95% solvent B; 22~27 min, 95% solvent B; 27.1~30 min, 5% solvent B. The sample injection volume was 5 µL and the flow rate was set to 0.3 mL/min. The column temperature was maintained at 40 °C. MS conditions: primary and secondary mass spectrum data were collected using Q Exactive mass spectrometer (Thermo Fisher Scientific, Waltham, MA, USA). The mass/nucleus ratio was 150–1500, the primary resolution was 70,000, the AGC was 1 × 10^6^, and the maximum injection time was 100 ms. According to the strength of parent ions, Top3 was selected for fragmentation, and secondary information was collected, with a secondary resolution of >35,000, AGC was 2 × 10^5^, the maximum injection time was 50 ms, and the stepped sequence was set to 20, 40, and 60 eV. Ion source (ESI) parameter settings: sheath gas flow rate was 40, Aux gas flow rate was 10, the positive ion mode of spray voltage(|KV|) was 3.80, and the negative ion mode was 3.20. The capillary temp was 320 °C and the aux gas heater temp is 350 °C. After mass spectrometry detection was completed, Compound Discoverer 3.0 (Thermo Fisher Scientific, USA) software was utilized for metabolic data processing. The metabolites of all samples were qualitatively and quantitatively analyzed by mass spectrometry combined with the BGI Library and the mzCloud database. Then, principal component analysis (PCA) was performed on the metabolites of Zhonghua6 and Yuanza 9102 seed samples to identify the overall metabolic differences among inoculated and comparable samples. Hierarchical cluster analysis (HCA) was conducted for the accumulation patterns of the same metabolites detected in different samples of Zhonghua6 and Yuanza9102 seeds using R-4.2.3 software (www.r-project.org, accessed on 15 March 2023). The KEGG database was searched for functional annotation and metabolic pathway analysis of the detected differential metabolites. The Python package was exploited to identify statistically significantly enriched pathways using Fisher’s exact test.

### 5.7. Joint Analysis of Transcriptome and Metabolome

In order to better comprehend the interaction between the transcriptome and the metabolome, DEGs and differential metabolites were mapped to the KEGG pathway database to gather information about their shared pathways. Then we screened for DEGs and the differential metabolites involved in the pathway and mapped the pathway profile.

## Figures and Tables

**Figure 1 toxins-15-00414-f001:**
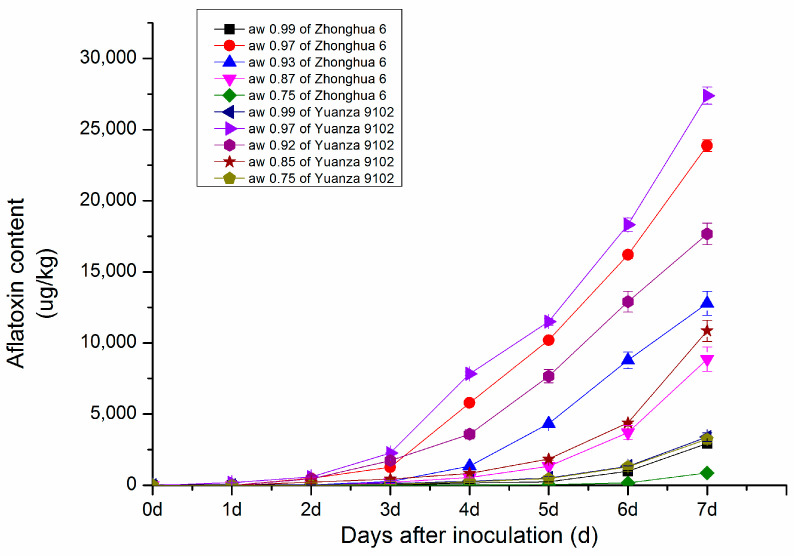
The dynamic changes in aflatoxin content in resistant genotype Zhonghua 6 and moderate susceptible Yuanza 9102.

**Figure 2 toxins-15-00414-f002:**
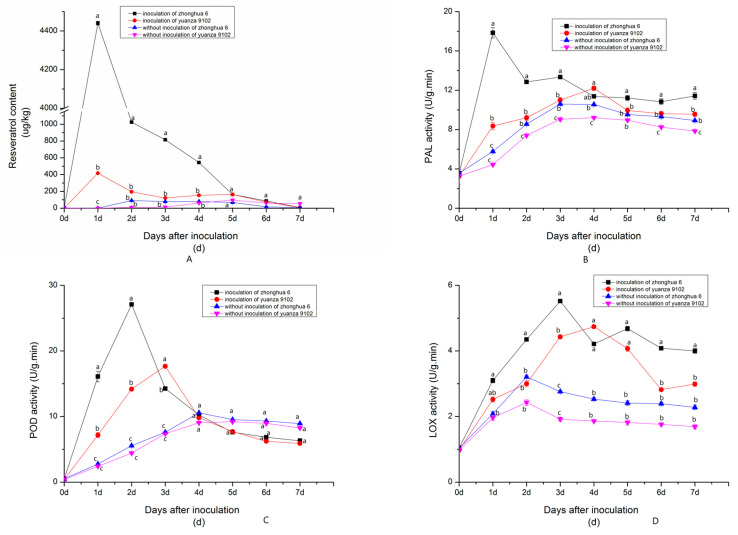
The dynamic changes in resveratrol and enzyme activity of two varieties peanut to *A. flavus* ((**A**): resveratrol content; (**B**): PAL activity; (**C**): POD activity; (**D**): LOX activity). Data are expressed as means ± standard deviations of triplicate assays. The different alphabetic superscripts in each period are significantly different (*p* < 0.05).

**Figure 3 toxins-15-00414-f003:**
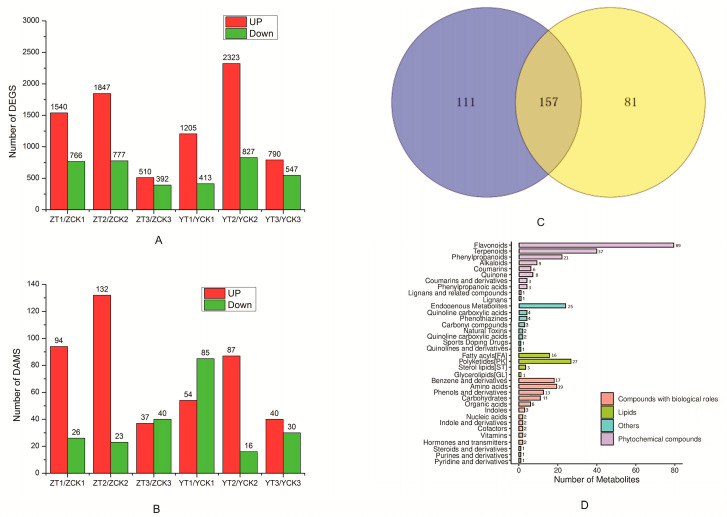
Quantity statistics and Venn diagram analysis of DEGS and DAMs. (**A**) Number of differentially expressed genes; (**B**) number of different metabolites; (**C**) Venn analysis of different metabolites; (**D**) classification of differential metabolites.

**Figure 4 toxins-15-00414-f004:**
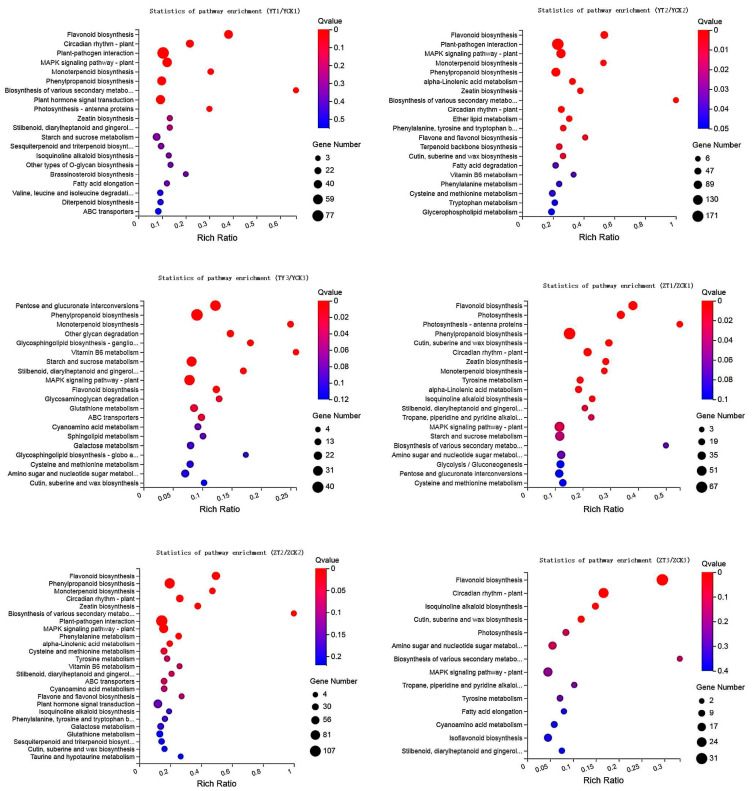
KEGG enrichment analysis of genes. The size of dots in the figure indicated the number.

**Figure 5 toxins-15-00414-f005:**
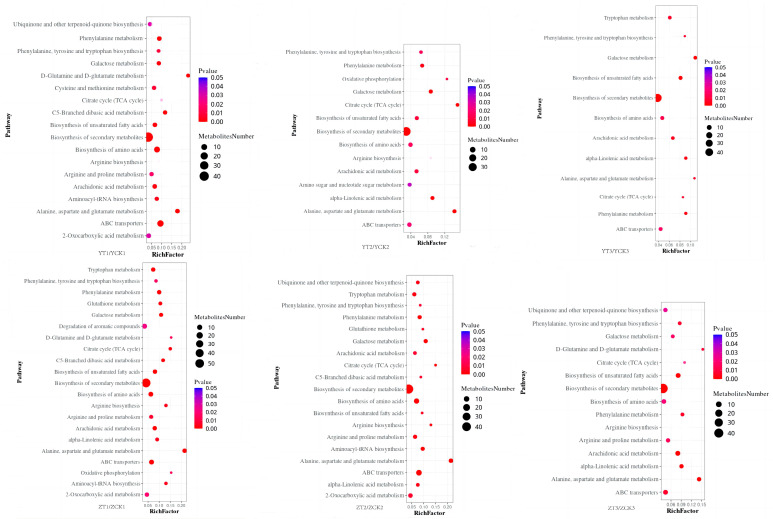
KEGG enrichment analysis of different metabolites. The size of dots in the figure indicated the number of genes in this pathway, and the color of dots corresponded to different q value ranges.

**Figure 6 toxins-15-00414-f006:**
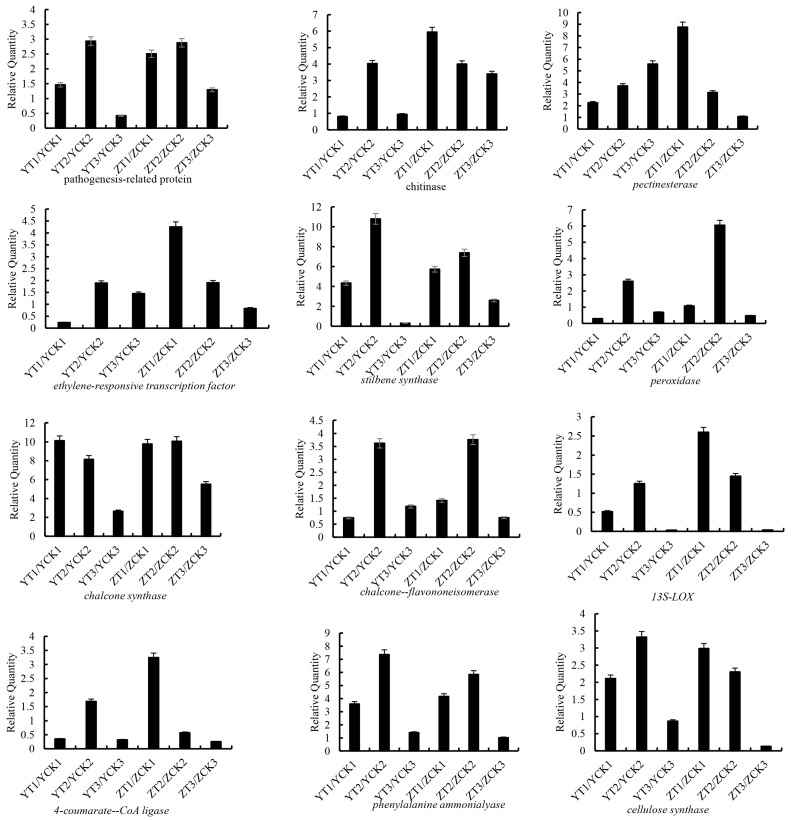
The relative expression levels of resistance related genes at different infection times. The reactions were performed in triplicate. Note: Data represent the means ± SD of three replicates.

**Figure 7 toxins-15-00414-f007:**
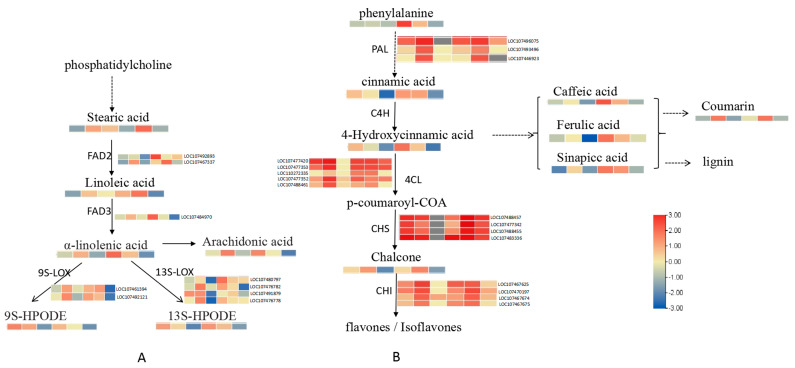
The integrated analysis of transcriptome data and metabolite analysis. (**A**): The dynamic change process of differentially expressed genes and differentially expressed metabolites in the α−linolenic acid metabolic pathway; (**B**): The dynamic change process of differentially expressed genes and differentially expressed metabolites in the phenylpropane metabolic pathway.

## Data Availability

The data that support the findings of this study are available from the corresponding author upon reasonable request.

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
