# Peer review of "Transcriptomic and Metabolomic Analyses of the Response of Resistant Peanut Seeds to Aspergillus flavus Infection"

_toxins, 2023, doi:10.3390/toxins15070414_

Round 1
Reviewer 1 Report
The manuscript requires an English language revision, in order to fix the number of inaccuracies present. But, overall, an extensive revision of the scientific writing style should be applied, as a lot of weird and unconventional sentences dot the main text. "there were significant differences in the biological processes involved in pathogen infection" is only an example: since we are looking at metabolites and transcription changes in a living organism, it's almost obvious that data refer to "biological processes"..... Authors must be less naive in their expressions.
The material and methods section is difficult to follow...so many inaccuracies and deep syntax/grammar errors are present that make hard to clearly reproduce the experiments. Enzyme assays are only cited when they must be -at least in brief - reported.
Personally, I didn't like how the discussion has been developed: I found many simple repetitions of presented data, that are not exactly discussed rather than briefly contextualized within the existent literature. This is my personal opinion, due to my preference toward a more comprehensive and "creative" explanation of results in their biological context....However, since the amount of experiments and provided data are so huge -as omic usually are-, I recognize that could be hard to produce a brilliant discussion.
Conclusions are not adequate to both the kind of research presented and the Journal audience; "All these results further confirmed that peanut infected by A. flavus can activate a variety of defence mechanisms, and the response to A. flavus is more rapid in resistant materials." is, once again, very naive: why there would be the necessity to confirm something that has been so widely demonstrated in so many plant species affected by pathogens? What does make THIS particular research so interesting and important to be shared?
Line 92: title "2.1. Peanut material and treatments" is not correct... Please check the relevant literature for a more appropriate title. Additionally, no mention is done about the kind of contenitor in which seeds were incubated, nor the presence of any wetted paper etc. More important: I can guess that seeds after 7 days were fully germinated:
Line 102: "Aspergillus flavus" must be in italic; 1*104 must be correctly formatted.
Lines 97-100: both syntax and grammar errors. Please fix the sentence.
Line 122: sentence "seeds from uninfected plants served as the control group" is quite obscure, as no "plants" were really infected, but seeds were superficially treated with A. flavus spores. What do Authors mean? Plantlets from non-infected seeds? Or something else? Moreover, three biological replicates are not sufficient for this kind of studies....
Line 124: RNA "purity" can not be " identified", but "checked".
Line 195: "A. flavus" must be in italic
Graphs: "cultural time" as axis label is not correct; change with "Time from inoculation" or "days after inoculation"
Figure 4A: horizontal axis' label is difficult to read;
Figure 4D: the salmon label "compounds with biological role" is highly incorrect as well as misleading.... what did Authors mean? Don't lipids or phytochemical compounds have any biological role???
Figure 7: which pathogenesis related protein? Please indicate; additionally, no statistical significance symbols are reported over the data.
References: "Park et al. 2005" is definitely NOT the right citation for "PR genes are generally not expressed....or other stimuli" sentence (line 413): this concept is such an important and general statement that requires more than one reference, but overall not just one so specialistic. Here, a couple of review papers on PRs in plants are needed.
Line 424: "plant resistance" change with "plant response"
Line 532: "the mechanism underlying the resistant of peanut to A. flavus infection" change in a less general sentence: Authors only investigated few varieties.
The Data Availability Statement" (The data that support the findings of this study are available from the corresponding author upon reasonable request.) should be a little bit reconsidered: there're not unreasonable requests, in science, for asking the access to bioinformatic or other metadata. This is the basis of scientific research.
An English language editing is required
Author Response
The material and methods section is difficult to follow...so many inaccuracies and deep syntax/grammar errors are present that make hard to clearly reproduce the experiments. Enzyme assays are only cited when they must be -at least in brief - reported.
Our response: The material and methods section have been modified in the manuscript
Personally, I didn't like how the discussion has been developed: I found many simple repetitions of presented data, that are not exactly discussed rather than briefly contextualized within the existent literature. This is my personal opinion, due to my preference toward a more comprehensive and "creative" explanation of results in their biological context....However, since the amount of experiments and provided data are so huge -as omic usually are-, I recognize that could be hard to produce a brilliant discussion.
Conclusions are not adequate to both the kind of research presented and the Journal audience; "All these results further confirmed that peanut infected by A. flavus can activate a variety of defence mechanisms, and the response to A. flavus is more rapid in resistant materials." is, once again, very naive: why there would be the necessity to confirm something that has been so widely demonstrated in so many plant species affected by pathogens? What does make THIS particular research so interesting and important to be shared?
Our response: we have modified the sentence.
Line 92: title "2.1. Peanut material and treatments" is not correct... Please check the relevant literature for a more appropriate title. Additionally, no mention is done about the kind of contenitor in which seeds were incubated, nor the presence of any wetted paper etc. More important: I can guess that seeds after 7 days were fully germinated:
Our response: Peanut material and treatments have been changed with Peanut material and pathogen inoculation . Additionally, the seeds were incubated as follows :At the same time, 20.0 g the shelled peanuts that were surface-sterilized for 45s by 75% ethanol and rinsed thrice with sterile distilled water were inoculated with 1 ml spores suspension (1*104 spores/mL) in a plantlet bottle. In the control, 1ml 0.05 % Tween-80 solution was placed to the peanut seeds. The inoculated samples and the control were placed in constant temperature incubator with 100% relative humidity at 30°C in dark. Samples which were taken continuously for 7 days after inoculation and immediately frozen in liquid nitrogen and stored in the -80℃. Each group was repeated for three times.
Line 102: "Aspergillus flavus" must be in italic; 1*104 must be correctly formatted.
Our response: we have modified the sentence.
Lines 97-100: both syntax and grammar errors. Please fix the sentence.
Our response: we have modified the sentence. Highly toxigenic strain A. favus 3.2890 from our lab was cultured on Potato Dextrose Agar for 7 days at 30◦C, Conidia were then collected and suspended in sterile water containing 0.05% Tween-80 with a concentration of 1*104 spores/ml and were stored at -70℃.
Line 122: sentence "seeds from uninfected plants served as the control group" is quite obscure, as no "plants" were really infected, but seeds were superficially treated with A. flavus spores. What do Authors mean? Plantlets from non-infected seeds? Or something else? Moreover, three biological replicates are not sufficient for this kind of studies....
Our response: I'm sorry I didn't describe it clearly, the sentence should describe as follows : The peanut seeds of Zhonghua 6 and Yuanza 9102 (inoculated with A. flavus and the control group) were collected at 1 dpi, 3 dpi and 7 dpi under the optimum water activity for RNA isolation.
Line 124: RNA "purity" can not be " identified", but "checked".
Our response: Identified have been changed with checked.
Line 195: "A. flavus" must be in italic
Our response: we have made modified in italic
Graphs: "cultural time" as axis label is not correct; change with "Time from inoculation" or "days after inoculation"
Our response: "days after inoculation" have been changed with "Time from inoculation"
Figure 4A: horizontal axis' label is difficult to read;
Our response: we have made adjustments
Figure 4D: the salmon label "compounds with biological role" is highly incorrect as well as misleading.... what did Authors mean? Don't lipids or phytochemical compounds have any biological role???
Our response: We classified the metabolites according to the different physical and chemical properties in KEGG and HMDB database, and made statistics of each type of metabolites in the way of bar graph. the metabolites in The four categories in the figure , If compared from the functional level, there is crossover between them.
Figure 7: which pathogenesis related protein? Please indicate; additionally, no statistical significance symbols are reported over the data.
Our response: tatistical significance symbols haved been reported over the data.
References: "Park et al. 2005" is definitely NOT the right citation for "PR genes are generally not expressed....or other stimuli" sentence (line 413): this concept is such an important and general statement that requires more than one reference, but overall not just one so specialistic. Here, a couple of review papers on PRs in plants are needed.
Our response: we have changed in the manuscript.
Line 424: "plant resistance" change with "plant response"
Our response: "plant resistance" have been changed with "plant response"
Line 532: "the mechanism underlying the resistant of peanut to A. flavus infection" change in a less general sentence: Authors only investigated few varieties.
Our response: Our description is not accurate. It has been modified to
The Data Availability Statement" (The data that support the findings of this study are available from the corresponding author upon reasonable request.) should be a little bit reconsidered: there're not unreasonable requests, in science, for asking the access to bioinformatic or other metadata. This is the basis of scientific research.
Our response: Your proposal is very reasonable, and we have revised it in the manuscript;
Data Availability Statement: Not applicable.
Reviewer 2 Report
Good research, just some minor revision required

Author Response
- We thank you for the critical comments and helpful suggestions. We have taken all these comments and suggestions into account, and have made major corrections in this revised manuscript.
-
We have modified in the manuscript.
We hope the revised manuscript is now acceptable to you. If not. We are glad to receive any further feedback which we shall continue to apply our best effort to address.
Reviewer 3 Report
The manuscript ID toxins-2409088 entitled " Transcriptomic and metabolomic analyses of the response of resistant peanut seeds to Aspergillus flavus infection", is an interesting study. But my suggest it needs to a major revision to be published in this journal. I do have some comments about the manuscript and data that could improve the overall quality of the manuscript:
1- Latin names must be written in italics ( for examples see lines 26,95,102,193 and etc….. ( Aspergillus flaves)).
2- I suggest putting figures of healthy and infected plants (peanut) as well as figures of the pathogen (Aspergillus flaves).
3- Abstract: Although carefully written, it needs to be enhanced with the promising results of the study (it lacks any quantitative data).
4- Line 24: This study does not discuss the production of peanuts, so I suggest delete this sentence: "Thus, these findings have important practical significance for peanut production".
5- Introduction:
· You have written in a wonderful way; however, I suggest adding a section about the mycotoxins and pathogen and how it infects the plant.( this references will support this section doi.org/10.1016/j.bcab.2019.101314 and doi.org/10.3390/jof8050482).
6- Materials and Methods:
· Line 95 : what is the source of pathogen ?
· Line 102: please check the number.
· Line 109: Please explain the method used in detail ( extraction and determination).
7- Results:
· Lines 211:213: ( this sentence must be removed to discussion section).
· Figures 3,4,5, and 6 need to high resolution.
8- Discussion: It was written in a scientifically correct manner.
9- Conclusion:
· Line 542: I suggest to remove this sentence ( Thus, these findings have important practical significance for peanut production).
10- Reference:
· There are references written incorrectly that should be rephrased ( see lines 48, 89, 406 and etc……).
· There are many references not correlated to the study, so I suggest replacing them with another references (such as references 26, 28, 26,28,29,30,31,39,40,41,43,46 and etc....)
Minor editing of English language required.
Author Response
- We thank you for the critical comments and helpful suggestions. We have taken all these comments and suggestions into account, and have made major corrections in this revised manuscript.
- Latin names must be written in italics ( for examples see lines 26,95,102,193 and etc….. ( Aspergillus flaves)).
Our response: Latin names have be written in italics in the manuscript
- I suggest putting figures of healthy and infected plants (peanut) as well as figures of the pathogen (Aspergillus flaves).
Our response: because the figure has a little blur,so I did not put in the manuscript.
- Abstract:Although carefully written, it needs to be enhanced with the promising results of the study (it lacks any quantitative data).
Our response: We have modified in the manuscript.
4- Line 24: This study does not discuss the production of peanuts, so I suggest delete this sentence: "Thus, these findings have important practical significance for peanut production".
Our response: We have remove this sentence: "Thus, these findings have important practical significance for peanut production".
5- Introduction:
- You have written in a wonderful way; however, I suggest adding a section about the mycotoxins and pathogen and how it infects the plant.( this references will support this section doi.org/10.1016/j.bcab.2019.101314 and doi.org/10.3390/jof8050482).
Our response: We We have modified in the manuscript.
6- Materials and Methods:
- Line 95 : what is the source of pathogen ?
- Line 102: please check the number.
- Line 109: Please explain the method used in detail ( extraction and determination).
Our response: Highly toxigenic strain A. favus 3.2890 from our lab and the concentration of Conidia is 1× 104 spores/ml. The extraction and determination of the Aflatoxin、Enzyme Activity and Resveratrol have been described detail in the manuscript.
7- Results:
- Lines 211:213: ( this sentence must be removed to discussion section).
- Figures 3,4,5, and 6 need to high resolution.
Our response: We have remove this sentence and the high resolutionof the figures have been changed.
- Discussion: It was written in a scientifically correct manner.
Our response: We have modified the discussion ,the following discussion have been added in the revised manuscript.
9- Conclusion:
Line 542: I suggest to remove this sentence ( Thus, these findings have important practical significance for peanut production).
Our response: We have remove this sentence
10- Reference:
- There are references written incorrectly that should be rephrased ( see lines 48, 89, 406 and etc……).
- There are many references not correlated to the study, so I suggest replacing them with another references (such as references 26, 28, 26,28,29,30,31,39,40,41,43,46 and etc....)
Our response: Some reference have been changed in the manuscript.
We hope the revised manuscript is now acceptable to you. If not. We are glad to receive any further feedback which we shall continue to apply our best effort to address.
Round 2
Reviewer 1 Report
The new version of the manuscript has been improved. However, references doi.org/10.1016/j.bcab.2019.101314 and doi.org/10.3390/jof8050482 are definitely NOT relevant to the description of plant-pathogen-mycotoxins relatioship. Author are STRONGLY advised to change with more focused and eminent literature.
Editing by Journal technical office would be able to fix minor issues
Author Response
Dear reviewer:
we really appreciate you for your carefulness and conscientiousness, Your suggestions are really valuable and helpful for revising and improving our paper. According to your suggestions, we have made the following revisions on this manuscript:
1.The new version of the manuscript has been improved. However, references doi. org/10. 1016/j.bcab.2019.101314 and doi.org/10.3390/jof8050482 are definitely NOT relevant to the description of plant-pathogen-mycotoxins relatioship. Author are STRONGLY advised to change with more focused and eminent literature.
Response:Thank you very much for your advice,we have changed references about doi.org/10.1016/j.bcab.2019.101314 and doi.org/10.3390/jof8050482 with other references. In the manuscript, we say “lots of biological strategies to battle contamination have been developed with limited success, So We changed the corresponding reference with the three reference(1. Investigation of Pseudomonas fluorescens strain 3JW1 on preventing and reducing aflatoxin contaminations in peanuts. 2. Biological control products for aflatoxin prevention in Italy: Commercial field evaluation of atoxigenic Aspergillus flavus active ingredients. 3. Non-aflatoxigenic Aspergillus flavus to prevent aflatoxin contamination in crops: advantages and limitations).
2.Minor editing of English language required
Response:The editing of English language has been made in the manuscript. The amendments are highlighted in red in the revised manuscript.

Reviewer 3 Report
We thank you for your efforts in improving the manuscript
Author Response
Dear reviewer:
- We thank you for the critical comments and helpful suggestions on our manuscript for the major Revisions. We have taken all these comments and suggestions into account, and have made major corrections in this revised manuscript.
- Thank you for reviewing my manuscript again.